# Absolute gravity measurements at Brest (France) between 1998 and 2022

| 3  |                                                                                                                                                  |
|----|--------------------------------------------------------------------------------------------------------------------------------------------------|
| 4  | Marie-Françoise Lalancette <sup>1</sup> , Guy Wöppelmann <sup>2</sup> , Sylvain Lucas <sup>1</sup> ,                                             |
| 5  | Roger Bayer <sup>3</sup> , Jean-Daniel Bernard <sup>4</sup> , Jean-Paul Boy <sup>5</sup> , Nicolas Florsch <sup>2</sup> , Jacques                |
| 6  | Hinderer <sup>5</sup> , Nicolas Le Moigne <sup>3</sup> , Muriel Llubes <sup>6</sup> , Bernard Luck <sup>4</sup> , and Didier Rouxel <sup>1</sup> |
| 7  |                                                                                                                                                  |
| 8  |                                                                                                                                                  |
| 9  | <sup>1</sup> Shom, CS 92803, 29228 Brest, France                                                                                                 |
| 10 | <sup>2</sup> LIENSs, CNRS – La Rochelle University, 17000 La Rochelle, France                                                                    |
| 11 | <sup>3</sup> Géosciences Montpellier, 34095 Montpellier, France                                                                                  |
| 12 | <sup>4</sup> EOST, CNRS – Université Strasbourg, 67084 Strasbourg, France                                                                        |
| 13 | <sup>5</sup> ITES, CNRS – Université Strasbourg - ENGEES, 67084 Strasbourg, France                                                               |
| 14 | <sup>6</sup> GET, CNRS – Université Paul Sabatier, 31400 Toulouse, France                                                                        |
| 15 |                                                                                                                                                  |
| 16 | Correspondence: G. Wöppelmann (guy.woppelmann@univ-lr.fr)                                                                                        |
| 17 |                                                                                                                                                  |
| 18 |                                                                                                                                                  |
| 19 |                                                                                                                                                  |
| 20 |                                                                                                                                                  |
| 21 | Keywords:                                                                                                                                        |
| 22 | Gravimeter; Gravity; Vertical land motion; Brittany                                                                                              |
| 23 |                                                                                                                                                  |
| 24 |                                                                                                                                                  |
| 25 | Short summary                                                                                                                                    |
| 26 | This study presents 25 years of carefully processed gravity measurements from western France, offering                                           |
| 27 | a unique dataset to support investigations of long-term land motion and sea level change. The data are                                           |
| 28 | consistent with satellite-based observations and are made available for use in future geophysical and                                            |
| 29 | climate-related research.                                                                                                                        |
| 30 |                                                                                                                                                  |

#### 32 Abstract

Repeated absolute gravity measurements, conducted once or twice per year, have proven valuable for quantifying slow vertical land motion with a precision better than 0.4  $\mu$ Gal per year (1  $\mu$ Gal = 10<sup>-8</sup> m 34 35  $s^{-2}$ ) after a decade or more. This precision is comparable to vertical velocity estimates derived from 36 continuously operating space-based geodetic techniques such as the Global Navigation Satellite System 37 (GNSS). Furthermore, absolute gravimeters are particularly well suited for long-term studies, as their 38 measurements are based on fundamental length and time standards (laser and atomic clock) and remain 39 independent of terrestrial reference frame realizations, unlike GNSS. Consequently, an absolute 40 gravimeter can return years or even decades later and provide relevant measurements, provided the 41 initial gravity data are well documented and the ground gravity marker remains undisturbed. Following 42 this line of thinking, we have compiled and consistently reprocessed absolute gravity measurements 43 collected between 1998 and 2022 in Brest, on the French Atlantic coast, near its century-long tide gauge 44 station. The entire dataset has been reanalyzed in accordance with international recognized standards 45 for instrumental and modelling corrections. This effort has yielded a 25-year time series of absolute 46 gravity values, which we present and document for future studies, along with details on our reprocessing 47 methodology. We assess the quality of this dataset and evaluate the extent to which the observed linear 48 gravity trend agrees with vertical velocity estimates from the nearby GNSS station co-located with the 49 tide gauge. The gravity data and metadata are made available via the French hydrographic agency Shom 50 portal (https://doi.org/10.17183/DATASET\_GRAVI\_BREST; Lalancette et al, 2024).

52

# 53 1 Introduction

Before the advent of precise satellite radar altimetry in the 1990s, tide gauges were the 55 primary source of sea level observations for scientific research. They still remain invaluable to 56 investigate climate-related changes over multi-decadal to century timescales (Pugh and Woodworth, 2014). The oldest sea level records available date back to the 17<sup>th</sup> century, of which 57 58 Brest is the longest instrumental series in France (Wöppelmann et al., 2006). A distinctive 59 feature of tide gauges is that they measure sea level with respect to the land upon which they 60 are grounded and thus record land level changes as well as sea level changes, which raises the 61 issue of separating solid Earth geophysical processes from ocean and climate-driven processes 62 in their records. A wide range of geophysical processes can result in land level changes (Emery 63 and Aubrey, 1991), but few have readily available models to correct the global tide gauge data set with a sub-millimetre per year uncertainty level (e.g., Glacial Isostatic Adjustment or GIA; 64 65 Tamisiea, 2011). An alternative approach to modelling is to measure the total land motion at a 66 tide gauge, irrespective of the underlying geophysical processes that affect land level.

3

The use of geodetic techniques to separate vertical land motion and changes in sea level at tide gauges was first reviewed by the International Association for the Physical Sciences of 68 69 the Oceans (IAPSO) within its Commission on Mean Sea Level and Tides (Carter et al., 1989), 70 and later on revisited as techniques and data analysis methods progressed (Carter, 1994; Neilan 71 et al., 1997; Blewitt et al., 2010; Wöppelmann & Marcos, 2016; Hamlington et al., 2020). 72 Following recommendations from such international groups, absolute gravity measurements 73 and Global Positioning System (GPS) - the first operational Global Navigation Satellite System 74 (GNSS) – started to be recorded at important tide gauges around the world in the early 1990s 75 (e.g., Baker, 1993; Zerbini et al., 1996). That is, shortly after transportable absolute gravimeters were available and able to address the challenging demand of 1-2  $\mu$ Gal (1  $\mu$ Gal = 10<sup>-8</sup> m·s<sup>-2</sup>) 76 77 precision (Niebauer et al., 1995), henceforth enabling to implement a systematic approach of 78 repeated observation campaigns at stations of interest (Faller et al., 2002).

Note that the above two types of instruments (absolute gravimeters and GNSS) provide 80 independent and complementary data: absolute gravity changes inform on mass variations and 81 vertical land motion, whereas GNSS can provide estimates of vertical land motion only 82 (Lambert et al., 2006). In particular, the role of GNSS has become dominant and the primary 83 method of choice due to its advantages in terms of cost, equipment installation and operating 84 ease, as well as positioning performances at the subcentimer precision level, ultimately yielding a substantial development of permanent GNSS stations (Blewitt et al., 2018). Nonetheless, 85 86 repeated absolute gravity measurements at tide gauges have proved worthwhile too, either as a 87 standalone technique (Williams et al., 2001) or in combination with GNSS, in particular to 88 overcome GNSS data analysis artifacts and potential systematic errors, such as those associated 89 with the alignment of GNSS positions and velocities with an international terrestrial reference 90 frame (Mazzotti et al., 2007; Teferle et al., 2009).

In France, the primary tide gauge for conducting absolute gravity measurements has 92 been the Brest one, having the longest sea level time series available in the Permanent Service 93 for Mean Sea Level (PSMSL) databank (Holgate et al., 2013). In addition, Brest station 94 contributes to the core network of tide gauges of the global sea level observing programme 95 under the auspices of the Intergovernmental Oceanographic Commission of UNESCO (IOC, 96 2012). The first absolute gravity campaigns at Brest were focused on investigating the ocean 97 tide loading in Brittany and understanding the environmental effects of proximity to the ocean 98 (Llubes et al., 2001). These initial objectives then shifted to the long-term monitoring of vertical 99 land motion at the Brest tide gauge.

4

With this paper, our goal is to describe the Brest absolute gravity station (Section 2), how the measurements were carried out (Section 3), what instruments and corrections were implemented, and how the measurements were reduced to a common reference (Section 4), ultimately yielding a consistent absolute gravity time series spanning circa 25 years, whose trend is estimated and compared to independent estimates of vertical land motion (Section 5), and whose data are hereby made available open and freely for future research (Section 6).

**2 Station setting**

A number of technical issues were carefully considered in the mid-1990s when planning 109 the site for absolute gravity measurements at Brest. The vicinity of the coastline to a gravity 110 site was known to be critical, impacting the quality of the gravity measurements and increasing 111 the variance of the data due to the microseismic noise from the nearby ocean waves (Baker, 112 1993). Fortunately, the problem can be greatly alleviated by locating the site a few kilometres inland from the coast. Carter et al. (1989) recommended establishing the absolute gravity sites 113 114 between 1 and 10 km inland from the tide gauges. Accordingly, the Brest absolute gravity site was established inland at 3.1 km from the tide gauge (Figure 1), similar to the Aberdeen 115 absolute gravity site in U.K., which is 3.2 km from the tide gauge (Williams et al., 2001). 116

The Brest absolute gravity measurements have been carried out above two ground floor 118 markers named Ref01 and Ref02 (Figure 1), located 10 meters apart from each other in the 119 basement of the building at the entrance of Shom, the French hydrographic agency. The height of Ref01 is 47.700 m above the national levelling datum known as NGF-IGN69 (Lucas, 2024). 120 121 In addition, the Ref01 marker was determined to be 0.010 mm below Ref02 using precise 122 levelling (Lucas, 2024). Interestingly, Wöppelmann et al. (2008) found six first-order levelling 123 surveys in the national mapping agency archives, which were carried out at regular intervals between 1889 and 1996. Their findings indicated local stability of the area up to 20 km 124 125 eastwards of Brest. Noteworthy, the height differences between the tide gauge benchmark and 126 a benchmark nearby the Shom absolute gravity site (designated as NO-1 and NO-5 in their 127 Table 1, respectively) did not exceed one millimetre over 73 years. That is, the Brest area appears stable well within the spirit levelling uncertainty level. Poitevin et al. (2019) further 128 129 confirmed the geodetic local stability of the Brest area using InSAR (Interferometry Synthetic Aperture Radar) data over the recent decades (between 1992 and 2000 with ERS-1/2 satellite 130 data, and between 2002 and 2008 with ENVISAT satellite data). 131

5

The observed geodetic stability of the Brest area is consistent with the geological setting 137 138 of a basement mainly composed of metamorphic crystalline rocks (Gneiss of Brest), which were 139 emplaced during the Cadomian (650-550 Myr) and Variscan (420-290 Myr) orogenesis 140 according to Cagnard (2008). Furthermore, Brest is located on a passive margin far from any 141 active zone of the European plate boundary. Interestingly, Brest may be part of the peripheral crustal bulge developed during the last glaciation (Emery and Aubrey, 1991). The bulge area 142 143 was once rising due to ice load of the British-Irish Ice Sheet (BIIS) and, after the deglaciation, 144 sinking. Presently, the central sector of the BIIS, broadly located on the deglaciated mountains 145 of Scotland, is undergoing an uplift (postglacial rebound) at a rate of about 1.6 mm/year, 146 whereas the surrounding areas (peripheral bulge) are subsiding at rates up to about 1.2 mm/year in southwest England (Shennan and Horton, 2002). However, Lenôtre et al. (1999) noted that 147 a slight error in the BIIS modelling (e.g., extent of glaciated area, history of deglaciation) can 148 149 result in a different position of Brest with respect to the peripheral bulge area.

# 151 **3 Instruments & data acquisition**

# 152 **3.1 Absolute & relative gravimeters**

Two absolute gravimeters of FG5 systems (Faller, 2002) were used to produce the time series presented in Section 5. These were manufactured by Micro-g Solutions (Niebauer et al., 154 155 1995) and numbered 206 and 228 (hereafter designated as FG5#206 and FG5#228). Briefly, 156 the FG5 absolute gravimeter measures the acceleration of a test mass (corner cube) in free fall 157 in a vacuum chamber by interferometry using a laser wavelength standard and an atomic frequency standard. The FG5 gravimeters are relatively cumbersome to operate in the field. 158 159 They originally weighed about 700 kg, but the ones used here had their electronic components miniaturised, yielding a weight reduction from 700 to 550 kg. Henceforth, they can effectively 160 161 be transported between stations in six boxes and have been successfully operated at remote sites as far as Antarctica (Amalvict et al., 2009). Figure 2 shows the FG5#228 operating above Ref01 162 163 ground floor marker in August 2007 and later above Ref02 marker in July 2022.

164