# Peer review of "Absolute gravity measurements at Brest (France) between 1998 and 2022"

_Earth System Science Data, 2025_

## Referee Comment (RC1)

**Review of**

**Absolute gravity measurements at Brest (France) between 1998 and 2022**

**authored by**

*Marie-Françoise Lalancette1, Guy Wöppelmann2, Sylvain Lucas1,
Roger Bayer3, Jean-Daniel Bernard4, Jean-Paul Boy5, Nicolas Florsch2, Jacques
Hinderer5, Nicolas Le Moigne3, Muriel Llubes6, Bernard Luck4, and Didier Rouxel*

**for ESSD-2025**

Summary of the article:

The authors present a time series of absolute gravimetric observations for the station Shom in the coastal city Brest. These 20 g-values have been measured over a period of 24 years, and are made available now for research application like investigations in sea level change or land uplift. The measured data have been prepared carefully (applying reductions for ocean and Earth's tides, atmospheric fluctuation, polar motion, etc., and editing) to obtain a homogeneous best data set for interdisciplinary research. Meta data and explaining descriptions are provided for future users. Due to the significance of the Brest station for the worldwide tide gauge network, the independent gravimetric series is of crucial importance w.r.t. the ongoing climate change.

Remarks:

I was a little unsure at the beginning because to the partly old references like Carter et al. (1989) or Baker (1993). But I know these old papers partly very well. They are in some way fundamental papers and worth to name them and refer to them. Overall, the reference list is a very good and proper list.

Starting from the beginning:

Line 71: Neilan et al., 1997. In References, I find 1998 and not 1997.

Line 82: I cannot find Lambert et al., 2006, in the references .

Line 159 and 160: It seems that the 700 kg are referred to the 2 FG5s. A single instrument has about half the weight. I would prefer here something like "One FG5 gravimeter weighed about 350 kg" to avoid any confusion.

Line 190, Table 1: I checked in the supplement the data for 01/08/2007, and I found the No sets with 188, but the drops per set are 50 and not 100. I have not checked all the other epochs.

Line 186: "microgal", not "microGal"

Line 186: "top-of-the-drop height". Some clear explanation would be helpful here. The top-of-the-drop is the resting position of the testmass at the vertical position z=0 (coordinate system of the data evaluation, positive downwards). The z=0 is important because when you shift the origin of the coordinate system to any other position, you will obtain different g-values. I often used the position of the first data pair (z,t) of the postprocessing as z=0 to avoid other problems.

Line 213: Hinderer and Luck 2005 ?

Line 218/219: "we edited the data . . .". What means "edited"? Elimination of gross errors?

Line 219: "best one-day gravity value (Table 1)". As I understand, Table 1 shows the mean g-values of the sets observed over more than one day, which corresponds with the supplement. Do you say here, that the g-values in Table 1 are the best one-day values? This here is confusing.

Line 256: Applying the evaluation software from the FG5 manufacturer, the gravity value from an FG5 is determined . . . .

Line 272: "The transfer of each absolute gravity value from the effective instrumental height (top-of-the-drop) to the common reference height . . .". The effective instrumental height is not the top-of-the-drop position. Here is the explaining extract from the paper of Wziontek et al. (2021):

[Figure]

**Fig. 1** Schematic representation of the effective position on the free-fall trajectory, where the determined $g$ is independent of the constant VGG $\gamma$ used within the observation equation of corner-cube gravimeters. The effective measurement height $h^*$ has its origin within the gravimeter itself (start of data acquisition) and depends on the processed section of the zero-crossings. The effective instrumental height $h_{ins}^*$ depends also on the setup of the gravimeter and has to be known to transfer the gravity value to a reference level (usually top of the benchmark) by using a VGG that can differ from $\gamma$.

Please read the caption of the figure above. Effec. Instr. Height is close to the 1/3 of the falling distance.

In References, I found 3 references not named in the text: line 488 (Boy and Hinderer2005, line 500 (De Linage 2003), line 574 (Van Camp and Vauterin, 2004).

Line 558: the year of Pugh and Woodworth? 2014?

My recommendation:

After small changes, the article will be ready for publication. The paper ensures sustainability for future investigations. Very good!

---

## Author Comment (AC1)

Review by Ludger Timmen with responses to the comments

**Summary of the article:**

The authors present a time series of absolute gravimetric observations for the station Shom in the coastal city Brest. These 20 g-values have been measured over a period of 24 years, and are made available now for research application like investigations in sea level change or land uplift. The measured data have been prepared carefully (applying reductions for ocean and Earth's tides, atmospheric fluctuation, polar motion, etc., and editing) to obtain a homogeneous best data set for interdisciplinary research. Meta data and explaining descriptions are provided for future users. Due to the significance of the Brest station for the worldwide tide gauge network, the independent gravimetric series is of crucial importance w.r.t. the ongoing climate change.

**Remarks:**

I was a little unsure at the beginning because to the partly old references like Carter et al. (1989) or Baker (1993). But I know these old papers partly very well. They are in some way fundamental papers and worth to name them and refer to them. Overall, the reference list is a very good and proper list.

*It is encouraging to see that the inclusion of these foundational works is recognized and valued. Thanks for this and the thoughtful remark on the reference list.*

Starting from the beginning:

Line 71: Neilan et al., 1997. In References, I find 1998 and not 1997.

*Indeed, the workshop took place in 1997, but the proceedings were published in 1998. We have corrected the citation year to 1998 accordingly.*

Line 82: I cannot find Lambert et al., 2006, in the references.

*Thank you for spotting this. We have now added the missing reference to the list.*

Line 159 and 160: It seems that the 700 kg are referred to the 2 FG5s. A single instrument has about half the weight. I would prefer here something like "One FG5 gravimeter weighed about 350 kg" to avoid any confusion.

*Thank you for the suggestion. We checked the weight from one of the transporter's invoices and confirmed that one FG5 gravimeter, packaged in its transport crates, weighed 247 kg. The manuscript is revised to read: "One FG5 gravimeter, packaged in its transport crates, weighed about 250 kg" to improve clarity.*

Line 190, Table 1: I checked in the supplement the data for 01/08/2007, and I found the No sets with 188, but the drops per set are 50 and not 100. I have not checked all the other epochs.

*Thank you for pointing this out. We have double-checked the supplementary project files and updated the statistics in Table 1 to ensure accuracy. In doing so, we identified and corrected four additional mistakes. The Excel supplementary file is also corrected accordingly.*

Line 186: "microgal", not "microGal".

*Thank you for spotting this. We have corrected it to use the lowercase "g" as recommended.*

Line 186: "top-of-the-drop height". Some clear explanation would be helpful here. The top-of-the-drop is the resting position of the testmass at the vertical position z=0 (coordinate system of the data evaluation, positive downwards). The z=0 is important because when you shift the origin of the coordinate system to any other position, you will obtain different g-values. I often used the position of the first data pair (z,t) of the postprocessing as z=0 to avoid other problems.".

*Thank you for pointing this out; it is indeed an important issue. The original sentence was too long and included multiple ideas, which reduced clarity. The updated text is inspired by your comment, as well as your related remark at Line 219 regarding the best one-day values. The revised version now reads:*

*"The gravity value of each observation campaign is also provided in Table 1 (col. 6) in microgal or µGal ($1\mu Gal = 10^{-8}\ m\cdot s^{-2}$) at the top-of-the-drop height above the floor marker ($g_0$ in Figure 3). In the FG5, this height corresponds to the resting position of the test mass (Figure 1 in Wziontek et al., 2021). Each gravity value in Table 1 (col. 6) is the average of the set gravity values over the given day (col. 1), with each set value itself being the average of the individual drops within that set."*

Line 213: Hinderer and Luck 2005 ?

*Thank you for noting this. We have added the publication year "2005" to the citation.*

Line 218/219: "we edited the data . . .". What means "edited"? Elimination of gross errors?

*By "edited," we mean that the data were carefully reviewed and, in cases where measurements extended over multiple days (two campaigns in 1999 and 2005), we retained the single full day of highest quality. Primarily, the discarded days were affected by poor weather conditions, which resulted in noticeably larger set scatter compared to the calmest day.*

Line 219: "best one-day gravity value (Table 1)". As I understand, Table 1 shows the mean g-values of the sets observed over more than one day, which corresponds with the supplement. Do you say here, that the g-values in Table 1 are the best one-day values? This here is confusing.

*As mentioned above, when data were collected over multiple days, we retained the highest-quality single full day of measurements. The g-values reported in Table 1 correspond to these one-day values, rather than to an average over all measurement days, which were affected by weather conditions. The text has been revised to clarify this. See some complementary information in our response to a similar comment from reviewer 2 with an illustration for the 2005 multi-day observation campaign.*

Line 256: Applying the evaluation software from the FG5 manufacturer, the gravity value from an FG5 is determined . . . .

*Thank you for the suggestion. We agree and have adopted the proposed wording in the revised manuscript.*

Line 272: "The transfer of each absolute gravity value from the effective instrumental height (top-of-the-drop) to the common reference height . . .". The effective instrumental height is not the top-of-the-drop position. Here is the explaining extract from the paper of Wziontek et al. (2021):

[Figure]

**Fig. 1** Schematic representation of the effective position on the free-fall trajectory, where the determined $g$ is independent of the constant VGG $\gamma$ used within the observation equation of corner-cube gravimeters. The effective measurement height $h^*$ has its origin within the gravimeter itself (start of data acquisition) and depends on the processed section of the zero-crossings. The effective instrumental height $h^*_{ins}$ depends also on the setup of the gravimeter and has to be known to transfer the gravity value to a reference level (usually top of the benchmark) by using a VGG that can differ from $\gamma$.

Please read the caption of the figure above. Effec. Instr. Height is close to the 1/3 of the falling distance.

*Thank you for this clarification and for pointing us to the relevant explanation in Wziontek et al. (2021). We agree that the effective instrumental height is not equivalent to the top-of-the-drop height. We have revised the sentence accordingly to avoid this confusion. The updated text now reads:*

*"The transfer of each absolute gravity value from the top-of-the-drop height ($h_{instr}$ in Figure 3) to the common reference height ($h_{ref}$ in Figure 3) was achieved using the actual vertical gravity gradients determined from measurements of relative gravity using a Scintrex CG3M or CG5."*

In References, I found 3 references not named in the text: line 488 (Boy and Hinderer2005, line 500 (De Linage 2003), line 574 (Van Camp and Vauterin, 2004).

*Thank you for pointing this out. These references were included in an earlier draft of the manuscript but are no longer relevant to the current content. We have removed them from the reference list.*

Line 558: the year of Pugh and Woodworth? 2014?

*Thank you for noting this. We have added the publication year "2014" to the reference.*

**My recommendation**:

After small changes, the article will be ready for publication. The paper ensures sustainability for future investigations. Very good!

*We sincerely thank the reviewer for his time and the positive feedback on our work. We appreciate the thoughtful suggestions and are glad that the contribution is seen as a valuable step toward supporting future investigations.*

---

## Author Comment (AC2)

Review by Hartmut Wziontek with responses to the comments

The contribution describes a 24 years long time series of absolute gravity measurements at Brest, France. The data itself have been already published with a DOI. It is an interesting record, in particular as leveling indicates an exceptional vertical stability of the region and the site is connected to the Brest tide gauge. The manuscript is informative and well written. However, a few points should be improved:

1) The stability of both FG5 gravimeters over the years need to be addressed more in detail. Comparisons of absolute gravimeters are mentioned (L329) but the actual results are missing, except for a general "offset dispersion" (L322) of absolute gravimeters. In particular the impact of the change of the dropping chamber of FG5-206 to FG5X-206 should be analyzed. Are there further stations, e.g. Larzac, that could be used to check the stability of both FG5?

*We thank the reviewer for pointing out the reference [4], which extends the comparisons in de Viron et al. (2011) and further confirms the stability of both FG5 gravimeters, especially when the two instruments were involved in the same comparison experiment. For instance, Table 5 of [4] shows that the difference between FG5#206 and FG5#228 is only 0.9 $\mu$Gal, i.e., well within the typical FG5 bias differences of 2.1 $\mu$Gal reported in [4]. We have revised the manuscript accordingly and added this reference.*

*Regarding FG5#228, this instrument benefits from repeated operation at its reference station in Montpellier (Larzac) whenever it is not engaged in field campaigns. Between 2014 and 2025, the daily repeatability of its gravity values at this site is 2.0 $\mu$Gal, with a daily set scatter of 1.2 $\mu$Gal (median of standard deviations). A previous reference station (2005-2012) showed slightly higher values (2.6 $\mu$Gal repeatability and 1.5 $\mu$Gal median set scatter), but this original station was destroyed due to new building construction.*

*As for FG5-206, the upgrade to FG5X-206 occurred in 2021, after our last campaign with this instrument in 2018. It therefore has no impact on the results reported here and remains outside the scope of the manuscript. However, to satisfy the reviewer's curiosity, we note that a comparison was conducted at Trappes in October 2022 between the upgraded FG5X-206 and FG5-228. The observed differences were 1.8 $\mu$Gal on pillar GR8, 1.2 $\mu$Gal on pillar GR40, and 3.1 $\mu$Gal on pillar GR29, the latter still within the error bar of the comparison (3.7 $\mu$Gal). These results suggest that the change of the dropping chamber did not introduce any statistically significant difference.*

2) I really don't understand the selection of measurements. Of course, if serious disturbances happened or the drop scatter exceeded a certain threshold a (part of) measurement should be excluded. Otherwise, an increased background noise should not affect the mean level, rather the precision of the measurement. So it is hard to understand why a significant part of the data was discarded, in order to meet the "best one-day gravity value" (L219). It would be worth to give more arguments for the data selection, e.g. an example where it becomes clearly visible why a significant part of the data was discarded.

*This is indeed an important point. We acknowledge that the cleaning and selection process involves a degree of subjectivity. It was based on data quality considerations and expert judgment, particularly the internal consistency of sets within each campaign. A*

*first example, discussed in the text (L212-216), is the January 2005 campaign. The figure below shows the gravity values for individual sets: data from January 18 and 19 exhibit a standard deviation of about 3.5 µGal, whereas January 20 shows a reduced scatter of 2.0 µGal. We retained only the highest quality sets from January 20.*

[Figure]

*A second case with more than one day of measurements is the October 1999 campaign. The daily results are summarized in the table below. Here too, weather conditions were poor early in the campaign and improved only toward the end. As such, the data from October 27, which showed both lower scatter and more stable conditions were retained.*

| Date | g (µgal) | Set scatter (µgal) |
|---|---|---|
| 21 October 1999 | 980 929 180.23 | 6.92 |
| 22 October 1999 | 980 929 181.27 | 6.14 |
| 25 October 1999 | 980 929 178.69 | 3.29 |
| 27 October 1999 | 980 929 177.32 | 2.97 |

*As a final remark, we note that apart from these two exceptional cases, the remaining gravity campaigns did not span multiple days. The manuscript has been modified to clarify this by adding: "In addition to the 2005 campaign, another exceptional case with multi-day measurements was in 1999, which was also affected by strong weather conditions." This addition also connects naturally with our recommendation to conduct campaigns over several days.*

3) I'm confused by the mix of instrumental heights: effective height and top-of-drop. It should be clearly discriminated that the effective height is different from top-of-drop (L273). In particular if the (applied) vertical gradients have changed over the years, the gravity value in the effective instrumental height would be the best choice to document the results. Also, it is advantageous to define a common reference height close to it (as specified in the caption of Fig. 5). Table 1 provides the gravity values at top-of-drop. If all measurements were evaluated with the gradient specified in chapter 4 it is not a serious problem. Nevertheless, I propose to

rework Table 1 to either use the effective instrumental height or a common reference height close to it, e.g. the values at 1.22 m used for Figure 5.

*A similar comment was made by Reviewer 1 regarding the need to clearly distinguish between the effective height and the top-of-drop height. We hope the revised version of the manuscript now clarifies this distinction and avoids confusion.*

*We confirm that the vertical gradients reported in chapter 4 have not changed over the years at a given location (Ref01 or Ref02).*

*The intention of Table 1 is to provide gravity values without applying the reduction to a common reference height using the vertical gravity gradients. Instead, these reductions are provided in the supplementary data file (Excel, sheet 2 "data"), which includes both the top-of-the-drop values and the ones reduced to the common reference height. The gradients and formulas are also included, allowing users to verify the reductions or apply them to a different height if needed.*

4) The site relocation is mentioned in the Conclusions with reference to Section 2. However, in Section 2, both sites are described but not the aspect of relocation. Please add some details why it was necessary to relocate and what might be the differences in the local conditions. Moreover, the trend essentially depends on this "jump", so the impact of the relative survey would be worth a critical review, also considering the overall stability documented by spirit leveling. Therefore, I suggest to additionally evaluate trends before and after changing the site.

*Thank you for this suggestion. The revised manuscript now includes an explanation of the rationale for relocating the gravity station from Ref01 to Ref02. This information has been added in Section 2, just after the sentence introducing the two markers. As detailed in the new text, the original room (Ref01) had become unsuitable due to recurrent maintenance work linked to a heating system, and a reduction in usable space caused by the expansion of the heating network. The adjacent room (Ref02) offers a more suitable environment: it is more spacious, better ventilated, and has generally lower and more stable temperatures, making it preferable for high-precision gravity measurements.*

*Regarding the trend, as the reviewer rightly noted, it appears to be influenced by the "jump" associated with the relocation. This impact is explicitly illustrated in Table 2 (Section 5.2.2), which presents two scenarios: one including Ref02 gravity values and one restricted to Ref01 only. The revised manuscript now further discusses Table 2 to highlight the role of the jump. We would also like to emphasize that our trend analysis is intended as an example of how the dataset could be used. The main purpose of publishing the complete data set is precisely to enable users to apply their own selection criteria (e.g., choice of time span, treatment of relocation, alternative modelling or analysis approaches) tailored to their scientific objectives.*

**Please find some minor comments below:**

L121: Is there really no significant height difference between Ref01 and Ref02? What is the uncertainty of the difference of 0.010 mm?

*Thank you for raising this point. The height difference between Ref02 and Ref01 comes from a levelling survey conducted in 2017 with an uncertainty of 0.001 m, typical for high-precision levelling over short distances. The revised manuscript now explicitly includes this uncertainty and reports the height difference as 0.009 ± 0.001 m (correcting the earlier mistake where it was incorrectly given as 0.01 mm). We also verified that this difference remained unchanged in a new levelling survey conducted in August 2025.*

L154: The FG5-206 was upgraded to FG5X. Please mention this, discuss the impact on the stability of the meter and also cite [1]

*The upgrade of FG5-206 to FG5X occurred in 2021, that is, after the last measurement reported here with FG5-206 (2018, see Table 1). Consequently, the upgrade has no impact on the results presented in this study and its discussion is out of the scope.*

L159: Please check the mass of the equipment. To my knowledge it is nowadays about 300-350 kg and was not more than 500 kg.

*A similar comment was raised by Reviewer 1. The manuscript was revised to read: "One FG5 gravimeter, packaged in its transport crates, weighed about 250 kg". (We checked the weight from the transporter's invoices).*

L186: Please define the unit microgal only once.

*Thanks for spotting this. Indeed, the unit was already defined at Line 76. We have now removed the repeated definition in parentheses.*

L196: You are referring to a very early paper by Carter et al. (1994). It is worth to be mentioned but is there indeed no more recent publications worth to be cited addressing the requirements to detect geodynamic trends, e.g. [2] or work for Fennoscandia?

*Thank you for this suggestion. The early paper by Carter et al. (1994) was instrumental in motivating the long-term monitoring at Brest. To our knowledge, the recommendation of conducting regular absolute gravity campaigns in that report has not been contradicted in the literature. If it is reaffirmed in more recent publications, such as reference [2], we are happy to cite them to strengthen the context, though we consider the original reference the most relevant for the Brest record.*

L207: You are only speculating about potential differences between both FG5. Please use comparison measurements to proof this (see my general comment 2) above)

*This point is addressed in our response to the general comment 2 (see above).*

L218: How can it be consistent to remove measurements just because of enlarged scatter of observations? (see comment 3) above)

*Please see our response to general comment 3 above.*

L246: You mention ocean tide loading up to 30 µGal. Is the tide model obtained from the CG3 record published? Otherwise, I suggest to publish it here in an appendix to allow application for future measurements or comparison with ocean tide models.

*Thank you for this comment. The value of 30 µGal cited in the manuscript is based on a visual estimate from Figure 2 in Llubes et al. (2001), which shows an FG5 record over several days, rather than from the CG3 record itself. A local ocean tide model derived from a CG3M record does exist, and the associated loading file is provided as supplementary material at the DOI landing page in the project files. The manuscript has been updated accordingly to clarify these points (end of Section 4.1).*

L272: This sentence is completely wrong. The effective instrumental height is not identical with top-of-drop! Please also check [3]. Was only the gradient changed or was the whole measurement reprocessed? If only the gradient was unified, then the gravity value should be first transferred with the original gradient to the effective instrumental height, and next with the actual gradient to the common reference height (or whatever reference is used).

*This point was also raised by Reviewer 1. We fully agree that the effective instrumental height is not equivalent to the top-of-the-drop height. The sentence has been revised to avoid this confusion.*

*All original measurements from the various campaigns were reprocessed using a uniform data analysis strategy, including consistent modelling, corrections, and application of the vertical gravity gradients reported in this study. This approach is stated at the beginning of Section 4.1 and is now reiterated at the end of that section to ensure clarity.*

L278: A reference height of 0 cm is given. It is impossible to measure directly on the floor or on the marker. Please correct.

*We thank the reviewer for pointing out this imprecise description. The manuscript has been corrected as follows: "Figure 4 illustrates how these measurements were performed using a dedicated, stable tripod with three predefined mounting levels (0 cm, 60 cm, and 120 cm above the floor), referring to the elevation of the instrument base."*

L278: What was the scatter of the individual gradient measurements? It would be interesting to document possible temporal changes.

*Thank you for this comment. The scatter of the individual gradient measurements was 0.06 µGal/cm at Ref01 and 0.04 µGal/cm at Ref02. No statistically significant temporal changes were observed, with estimated trends of +0.05 ± 0.12 µGal/cm/year at Ref01 and −0.08 ± 0.18 µGal/cm/year at Ref02. The revised manuscript has been updated to include these statistics.*

L319: Could you please also check the more recent paper [4] and the respective comparison reports to evaluate the stability of both FG5? See my general comment 2) above.

*We thank the reviewer for pointing out this reference. We have considered it and added to address the general comment 2 above.*

L354: The list of instrumental errors contributing to the uncertainty budget seems not applicable to absolute gravimeter in all cases: There is no "difference in vacuum condition". If

the vacuum is insufficient, no measurement can be done. Also, there is no phase response/transfer function known to me that influences the absolute gravity measurement, apart from non-linearity in the fringe detection. Please update/further elaborate and provide references.

*We thank the reviewer for pointing out these inaccuracies. We agree that "differences in vacuum condition" and "phase response/transfer function" are not appropriate error sources in this context. These have been removed from the manuscript, and the sentence has been revised to retain only the relevant remaining contributions.*

Section 5.2.1/Figure 6: If the measurements after the 2016 would be neglected, would the trend still be significant? Please explain how the time dependent error estimate was calculated.

*We thank the reviewer for this comment. As shown in Table 2, when the measurements after 2016 are excluded, the estimated trend is -0.13 ± 0.60 mm/year, which is not statistically significant. The trend was computed using a weighted least squares fit, with weights based on the variances of the individual data points. We assumed normally distributed and statistically independent measurement errors. This point has been clarified in the revised manuscript.*

Code availability: For the ETERNA software, the repository [5] at KIT should be cited instead of the link to the GFZ publication.

*Thank you for the suggestion. The link has been updated in the revised manuscript to cite the ETERNA repository at KIT as recommended.*

**References**

[1] T M Niebauer et al 2011 Metrologia 48 154, doi:10.1088/0026-1394/48/3/009

[2] Pálinkáš et al Acta Geodyn. Geomater., Vol. 7, No. 1 (157), 61–69, 2010

[3] Pálinkáš et al 2012 Metrologia 49 552 doi 10.1088/0026-1394/49/4/552

[4] Pálinkáš et al J Geod 95, 21 (2021), https://doi.org/10.1007/s00190-020-01435-y

[5] https://publikationen.bibliothek.kit.edu/1000151532 DOI: 10.35097/746